# Longitudinal causal effect of modified creatinine index on all-cause mortality in patients with end-stage renal disease: Accounting for time-varying confounders using G-estimation

**Mohammad Aryaie[1], Hamid Sharifi[2], Azadeh Saber[3], Farzaneh Salehi[4], Mahyar Etminan[5], Maryam Nazemipour[6], Mohammad Ali Mansournia[6]***

**1** Department of Epidemiology, School of Health, Shiraz University of Medical Sciences, Shiraz, Iran, **2** HIV/STI Surveillance Research Center, and WHO Collaborating Centre for HIV Surveillance, Institute for Futures Studies in Health, Kerman University of Medical Sciences, Kerman, Iran, **3** Physiology Research Center, Institute of Neuropharmacology, Kerman University of Medical Sciences, Kerman, Iran, **4** Department of Critical Care Nursing, Faculty of Nursing and Midwifery, Kerman University of Medical Sciences, Kerman, Iran, **5** Department of Ophthalmology, Medicine and Pharmacology, University of British Columbia, Vancouver, Canada, **6** Department of Epidemiology and Biostatistics, School of Public Health, Tehran University of Medical Sciences, Tehran, Iran

* mansournia_ma@yahoo.com, mansournia_ma@sina.tums.ac.ir

**Data Availability Statement:** Data are available from the Ethics Committee for researchers who

## Abstract

### Background

Standard regression modeling may cause biased effect estimates in the presence of time-varying confounders affected by prior exposure. This study aimed to quantify the relationship between declining in modified creatinine index (MCI), as a surrogate marker of lean body mass, and mortality among end stage renal disease (ESRD) patients using G-estimation accounting appropriately for time-varying confounders.

### Methods

A retrospective cohort of all registered ESRD patients (n = 553) was constructed over 8 years from 2011 to 2019, from 3 hemodialysis centers at Kerman, southeast of Iran. According to changes in MCI, patients were dichotomized to either the decline group or no-decline group. Subsequently the effect of interest was estimated using G-estimation and compared with accelerated failure time (AFT) Weibull models using two modelling strategies.

### Results

Standard models demonstrated survival time ratios of 0.91 (95% confidence interval [95% CI]: 0.64 to 1.28) and 0.84 (95% CI: 0.58 to 1.23) in patients in the decline MCI group compared to those in no-decline MCI group. This effect was demonstrated to be 0.57 (-95% CI: 0.21 to 0.81) using G-estimation.

meet the criteria for access to confidential data. Data access requests may be sent to the Vice-Chancellor for Research & Technology at "vcr@kmu.ac.ir" or to the Research and Technology Office at "kmu_research@yahoo.com.

**Funding:** The author(s) received no specific funding for this work.

**Competing interests:** NO authors have competing interests.

## Conclusion

Declining in MCI increases mortality in patients with ESRD using G-estimation, while the AFT standard models yield biased effect estimate toward the null.

## Introduction

Obesity, as measured by body mass index (BMI), is a major cause of death in the general population [1, 2]. However, obesity could increase longevity in patients receiving maintenance hemodialysis, known as "reverse epidemiology" of obesity or "obesity paradox" [3, 4], which recently has bred an ongoing debate as to whether such findings are plausible or applicable in everyday practice [5, 6].

Even if this inverse paradoxical association is postulated to be robust, as demonstrated using a marginal structural causal model appropriately accounting for time-varying confounders [7], BMI is unable to differentiate between lean body mass and fat mass [8]. The latter is more associated with inflammation, leading the mortality predictability of BMI ambiguous in patients on hemodialysis. In fact, lean body mass could better reveal changes in body mass than BMI over time so that lean body mass deterioration has been recently shown to be more strongly associated with mortality than declining BMI in patients on hemodialysis [9]. Furthermore, muscle mass, defined by creatinine-index level, and its change over time was recommended to be regularly measured for the nutritional assessment [10, 11].

Modified creatinine index (MCI), determined by sex, age, pre-dialysis serum creatinine, and single-pool Kt/V (spKt/V), has been introduced as a reliable, valid and simple surrogate marker of lean body mass [12, 13]. The effect of this time-varying index on all-cause mortality has been examined using standard regression models, e.g. time-dependent Cox regression model [9, 13, 14]; however, these models fail to provide unbiased effect estimates in the presence of time-varying confounders affected by prior components of time-varying exposure [15, 16]. For example, when the effect of receiving adequate dietary protein intake is of interest, inflammation which may suppress appetite [17, 18] is a time-varying confounder for hemodialysis patients' death. Thus receiving a diagnosis of inflammation may modify diet [19]. Moreover, the risk of inflammation might be affected by patients' earlier diet history [20, 21]. A substantial difference between effect estimates of causal and traditional models has been recently shown by Aryaie et al (22).

To overcome this problem, we used G-estimation of a structural accelerated failure time model (SAFTM), which could appropriately account for such time-varying variables that can at times act as both mediators and confounder [22, 23], to assess the effect of declining MCI on 8-year risk of all-cause mortality in patients with end-stage renal disease (ESRD). Results of this causal model were compared to those generated by standard time-varying accelerated failure time (AFT) Weibull model.

## Methods and materials

### Study population and follow-up

A retrospective cohort of all registered ESRD incident subjects, thrice-weekly received maintenance hemodialysis, (n = 568) aged ≥ 18 years was constructed from March 21, 2011, at Kerman, southeast of Iran. The follow-up ended at the time of death, transplantation, loss to follow-up, or administrative end of follow-up on December 23, 2019, whichever came first.

The research was approved by the ethical committee of Kerman university of medical science and three hemodialysis centers, including Shafa, Javadalaemeh, and Samenalhojaj centers (IR. KMU.REC. 1398,467; Reg. No. 97001038). According to the retrospective nature of this study, the informed consent was waived by the mentioned ethical committee. Moreover, all procedures were performed in accordance with relevant guidelines and regulations.

## Exposure, potential confounders and outcome

The modified creatinine index (mg/kg per day) was assessed at all visits (0 to 34 with 3-month intervals) using the following equation:

$$MCI = 16.21 + 1.12 \times [1 \; if \; male; 0 \; if \; female] - 0.06 \times age \; (years) - 0.08 \times single \; pool \frac{Kt}{V}$$
$$+ 0.009 \times serum \; creatinine \; before \; dialysis \; (\mu mol/L)$$

MCI level determined, by sex, age, pre-dialysis serum creatinine, and single-pool Kt/V (spKt/V), as a reliable, valid, and simple surrogate marker of lean body mass, like other studies [12, 13]. Then according to changes in MCI in each visit compared to the previous visit, patients were dichotomized to either the decline group or no-decline group. Based on expert opinion of a panel of nephrologists and epidemiologists, data on time-varying confounders were collected at all visits (0 to 34 with 3-month intervals) included body mass index (BMI), serum albumin, ferritin, white blood cell (WBC) count, and C-reactive protein (CRP). Time-fixed or baseline confounders included sex, age, and comorbidities listed in Table 1. A restricted cubic regression spline with four knots at the 5th, 35th, 65th, and 95th percentiles was used for ferritin and age. Data on potential confounders were collected from patient's routine clinical records. Laboratory values of creatinine and hemoglobin were measured monthly;

**Table 1. Baseline characteristics of patients with ESRD based on MCI levels, Kerman, Iran, 2011–2019.**

| | | Baseline exposure (MCI) status | | Outcome status | |
| --- | --- | --- | --- | --- | --- |
| | | Decline group (297) | No-decline group (256) | Death (168) | Alive (385) |
| | | No. (%) | No. (%) | No. (%) | No. (%) |
| **Demographic** | Sex (female) | 125(42.6) | 94(36.7) | 59 (35.1) | 162(42.0) |
| | Age (years) | 58.5 (14.6)[a] | 59.2 (15.2)[a] | 62.9 (12.9)[a] | 58.6 (14.65)[a] |
| | BMI[c] | 23.9 (4.3)[a] | 24.1 (4.4)[a] | 23.7 (4.0) [a] | 25.1 (4.9) [a] |
| **Comorbidities** | Diabetes | 193(65.8) | 169(66) | 119 (70.8) | 245 (63.6) |
| | Hypertension | 261(89) | 208(81.2) | 146 (86.9) | 325 (84.4) |
| | Cardiovascular disease | 61(20.8) | 48(18.7) | 48 (28.5) | 61 (15.8) |
| | hyperlipidemia | 21(7.11) | 16(6.2) | 11 (6.5) | 26 (6.7) |
| | Respiratory disease | 10(3.4) | 6(2.3) | 8 (4.7) | 8 (2) |
| | Cancer | 4(1.3) | 3(1.1) | 4 (2.3) | 4 (1.0) |
| | Laboratory tests | | | | |
| | CRP (positive) | 47(16) | 42(16.4) | 56 (33.3) | 39 (10.1) |
| | Albumin (g/dl) | 3.9 (0.5)[a] | 3.9 (0.5)[a] | 3.7 (0.4)[a] | 3.9 (0.5)[a] |
| | Ferritin (ng/ml) | 250 (132–364)[b] | 243 (127–374)[b] | 303 (170–546)[b] | 226 (105–355)[b] |
| | WBC (1000/μl) | 6.3 (1.6)[a] | 6.2 (1.5)[a] | 6.1 (1.6)[a] | 5.7 (1.3)[a] |

[a] mean (SD)

[b] median (IQR)

[c] defined as weight (kg)/height (m$^2$)

BMI: body mass index; CRP: C-reactive protein; WBC: white blood cell

serum albumin, ferritin and CRP were measured quarterly by standardized and automated methods. BMI was measured using dry weight within 5–15 min after hemodialysis session. After exclusion of subjects with missing information at baseline, 553 data on ESRD patients remained in the analysis, and all-cause mortality was considered as the main outcome, obtained from hospital information system registry.

## Causal diagram

Fig 1 is a causal diagram for the effect of MCI on all-cause mortality among ESRD patients. A(t) indicates MCI status at visit t, and Y(t) stands for death during the follow-up (visit t-1, visit t). L(t) consists a vector of measured time-varying confounders (e.g., BMI and ferritin) at visit t and L(0) includes time-fixed confounders (e.g., marital status, and diabetes) as well as the baseline values of time-varying confounders. Moreover, U(t) indicates all unmeasured risk factors for Y(t+1) such as residual kidney function. C(t) shows censoring (1:Yes, 0:No) during the period (visit t-1, visit t). The square around C(k) denotes our analyses are restricted to uncensored individuals. No arrows from U(t) to A(t) and C(t+1) assumes no selection bias due to unmeasured risk factors conditional on L(t). Causal diagrams have been described in details elsewhere [24–32].

## Statistical methods

**Standard models.** To estimate the association between time-varying MCI and all-cause mortality, accelerated failure time (AFT) Weibull models were used through two modeling strategies: in the first model, time-varying MCI was adjusted for time-fixed confounders including sex, age, comorbidities, and the baseline values of time-varying confounders. The second model was adjusted for time-varying confounders including albumin, CRP, ferritin, WBC count, and BMI plus all confounders adjusted in the first model. The implications of adjusting for baseline exposure and confounders in the longitudinal causal and regression

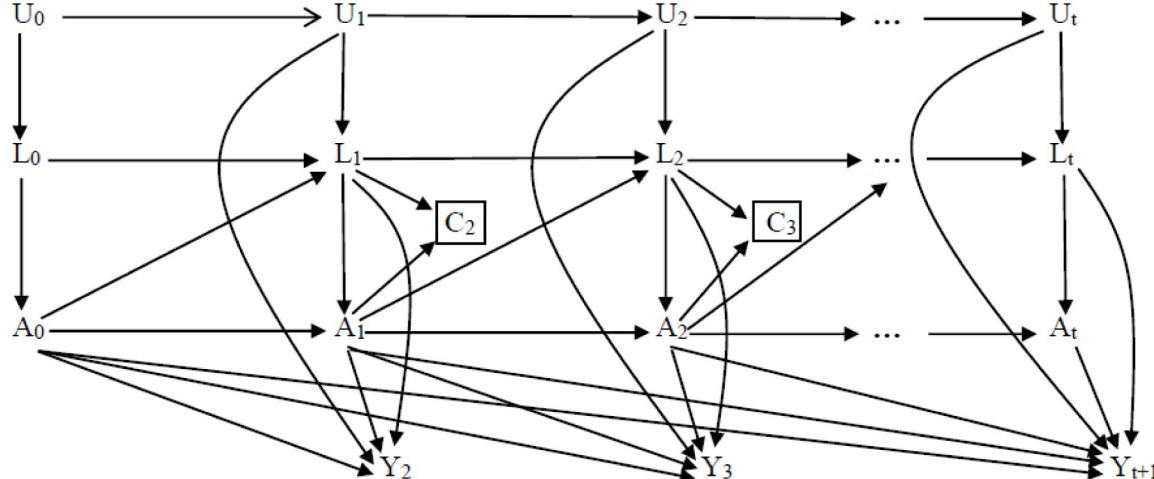

**Fig 1. Assumed causal diagram for the effect of lean body mass (A) on all-cause mortality (Y) among ESRD patients.** Note: Standard models are subject to two biases: over-adjustment bias (e.g., conditioning on $L_2$ blocks the indirect effect of $A_1$ on $Y_3$ through $L_2$), this bias occurs because $L_2$ is a time-varying confounder affected by the exposure A1 as well as an unmeasured causal risk factors $U_2$, and collider bias (e.g., conditioning on $L_2$ is common effect of $A_1$ and $A_2$. So, conditioning on $L_2$ associate $A_1$ and $U_2$, making $A_1$ a non-causal risk factor $Y_3$), this bias occurs because $L_2$ is a time-varying confounder affected by prior exposure $A_1$. But G-estimation appropriately account for such time-varying variables that can at times act as both mediators and confounders.

models have been explained elsewhere. [33]. Log-minus-log survival plots were also used to assessed Weibull, proportional hazards, and AFT assumptions.

**G-estimation.** G-estimation is a 2-stage iterative process: in the first stage, SAFTM links the causal parameter (effect of MCI on all-cause mortality) with the counterfactual survival time if individuals had never been exposed throughout the follow-up; in the second stage, the probability of MCIat each visit is modeled as a function of prior exposure and confounders history and counterfactual survival time using pooled logistic regression model [22, 34, 35].

In fact, G-estimation emulates a nested target trial in which exposure is randomly assigned at each visit t within strata of previous exposure and confounders [36]. This approach searches for the causal parameter of interest for which the counterfactual survival time would be independent of the exposure under the assumptions of well-defined exposure, conditional exchangeability, no measurement error, and correct model specification [37]. Moreover, to adjust for the potential selection bias [38–40] due to censored event (transplantation) and losses to follow-up in our study, the contribution of each individual was also weighted using inverse probability-of-censoring [25] in the process of G-estimation as follows:

For each individual, the visit-specific probability of being censored given prior exposure and confounders was estimated using pooled logistic regression to determine the conditional probability of remaining uncensored until the last visit. Next, the inverse of these subject-specific probabilities was used as weights to produce a pseudo-population in which nobody is censored, meaning that censored individuals were replaced with uncensored individuals with the same values of the exposure and confounders history. A mean weight of one would be necessary for correct model specification [25]. Then G-estimation was applied to the pseudo-population. Furthermore, G-estimation addresses administrative censoring to avoid selection bias by censoring individuals who survive until the end of follow-up as well as those who had an event and would have extended their counterfactual survival time beyond the end of follow-up if they had different exposure values than they actually had [41]. Finally, the 95% conservative confidence limits were obtained by finding a set of values of the causal parameter of interest that result in a P-value greater than 0.05 for the G-test of the hypothesis of no association between exposure and counterfactual survival time in the pooled logistic regression model [23]. The visit after baseline (second visit) was considered as the start of all analyses, performed using Stata version 14 (Stata Corp, College Station, Texas) [42].

## Results

Out of 568 patients with ESRD, 15 (2.6%) subjects with missing data at baseline or visit 1 were excluded. As a result, 553 ESRD patients were included in the study; 24 (4.3%) patients were censored during the follow-up: 4 due to loss to follow-up and 20 due to transplantation. There were 297 patients in decline MCI group and 256 patients in no-declined MCI group. During 8.8 years of follow-up, a total of 1492 person-years were followed in which 168 deaths occurred. The mortality rate was 113 per 1000 person-years (95% confidence interval [95% CI]: 97 to 131).

The baseline characteristics of patients according to MCI status have been illustrated in Table 1. The mean (SD) age was 59.7 (14.3) and 60.9% were male. Subjects in decline MCI group were more likely to have hypertension, hyperlipidemia, and cardiovascular and respiratory diseases, and had higher ferritin and WBC count compared with subjects in no-decline MCI group.

Survival time ratio and hazard ratio estimates using G-estimation of SAFTM and time-dependent AFT Weibull model are presented in Table 2. G-estimation of SAFTM yielded survival time ratio of 0.57 (95% CI: 0.21 to 0.81) in subjects who would have been always in

**Table 2. The effect estimates of MCI on mortality risk in patients with ESRD using AFT Weibull regression models and G-estimation of SAFTM, Kerman, Iran, 2011–2019.**

|  | Survival time ratio (95% CI) | Hazard ratio (95% CI) |
|---|---|---|
| Time-dependent AFT Weibull regression[a] | 0.91 (0.64, 1.28) | 1.08 (0.79, 1.48) |
| Time-dependent AFT Weibull regression[b] | 0.84 (0.58, 1.23) | 1.15 (0.83, 1.61) |
| G-estimation of SAFTM[b] | 0.57 (0.21, 0.81) | 1.62 (1.19, 3.91) |

CI, confidence interval.

[a]Adjusted for time-fixed confounders including sex, age, comorbidities, and baseline values of time-varying confounders.

[b]Adjusted for time-varying confounders including albumin, C-reactive protein, ferritin, white blood cell, and body mass index plus all above-mentioned confounders.

decline MCI group compared to those who would have been always in no-decline MCI group throughout the follow-up, whereas survival time ratios were 0.84 (95% CI: 0.58 to 1.23) using the second standard time-dependent AFT Weibull model (adjusting for both time-fixed and time-varying confounders), and 0.91 (95% CI: 0.64 to 1.28) using the first standard time-dependent AFT Weibull model (adjusting for time-fixed confounders and the baseline values of time-varying confounders).

The hazard ratio estimates (95% CIs) obtained by G-estimation, the second standard time-dependent AFT Weibull model, and the first standard time-dependent AFT Weibull model were 1.62 (1.19 to 3.91), 1.15 (0.83 to 1.61), and 1.08 (0.79 to 1.48), respectively. The mean (SD) of stabilized inverse probability-of-censoring weights was 1.00 (0.27).

## Discussion

The current study assessed the longitudinal causal effect of MCI on all-cause mortality using G-estimation, and compared the results with those estimated by standard models. The results showed declining MCI decreases time to mortality by 9% and 16% in the first and second standard model, respectively, whereas it was 43% based on G-estimation. Despite publication of several cohort studies [9, 13, 43, 44] attempting to estimate the association of lean body mass with all-cause mortality, no previous study has appropriately accounted for covariates which can concurrently act as both confounder and intermediate variables.

The findings of our standard models do not support previous studies' findings of a positive association between low muscle mass and mortality [9, 13, 45–48]. Compared with G-estimation, we showed that the effect of MCI on all-cause mortality tend to be biased and the survival benefit of no-decline in MCI was approximately 30% attenuated. Assuming that MCI may be associated with factors like serum albumin, C-reactive protein, and ferritin through common unmeasured causes such as deprived early-life conditions and poor diet intake, our standard models estimates may be substantially biased so that factors such as serum albumin and C-reactive protein (as markers of or associated with inflammation) are affected by low MCI values (as an indicator of poor muscle nutritional status) and go on to effect successive MCI changes, e.g. inflammation may modify diet. We adjusted for these confounders at baseline to address the exposure-confounder feedback that might have occurred prior to the study baseline in the first model, but did not account any further effect of MCI on these confounders, or these confounders on MCI over the study period. Moreover, we adjusted for the updated values of these confounders after baseline in the second model; nonetheless, this model does not take into account the fact that these confounders are affected by MCI, generating collider-stratification and over-

adjustment biases [15, 27, 49]. The direction and magnitude of the induced bias is unpredictable without adequate knowledge of error structures [25], and this might be the reason of attenuated association of MCI with mortality in other studies [9, 13] using standard models adjusting for almost the same confounders alike ours.

Only our G-estimation method correctly estimated the effect of hypothetical regimen of maintaining no-decline in MCI group compared with always decline on all-cause mortality. This causal model adjusts for confounding effect of time-varying confounders affected by prior exposure without introducing collider-stratification and over-adjustment biases. Only the results of G-estimation underscore the survival benefit of MCI as an indicator of lean body mass, and support generalizability of MCI to use in the skeletal muscle nutritional management for different populations receiving dialysis [9, 13, 45–48]. Declining lean body mass, such as that defined by MCI, is associated with the vital prognosis of hemodialysis patients [7, 50], and its decreasing trend over time may reflect poor nutritional status and is associated with physical frailty and poor prognosis including higher mortality [11, 13]. Therefore, it is recommended that MIC and its changes are measured regularly for the risk stratification or intervention to prevent the harmful effect of lean body mass declining, or provide the relative advantage of lean body mass increasing [10, 11].

The observed effect of MCI on mortality may be affected by different factors such as inflammation, poor dietary nutrition, hypercatabolism, and uremic toxins [51, 52]. However, compared with serum albumin, which is more commonly used as a surrogate nutritional indicator [53], MCI is a more specific and relatively stable index of somatic protein store [45, 54] with the advantage of being measured typically monthly; in contrast, the latter is collected less frequently by some dialysis facilities [13]. Moreover, while Vernaglion et al. [55] indicated that creatinine metabolism is not affected by inflammatory acute phase response, serum albumin is influenced by inflammation and also fluid status, rendering it a composite indicator [56]. It is important to note that some researches have shown MCI decreases at the same time prior to death as nutritional indices, including normalized protein catabolic rate, serum albumin, phosphate, and creatinine [57, 58].Thus, based on earlier findings [9, 12, 13] and the current study, MCI appears to be a valuable and easy access marker of lean body mass and deserves monitoring its change over time, which facilitates early detection of muscle wasting or sarcopenia trends, and offers intervention opportunities to stop, delay, or even reverse such harmful effect.

The validity of inference from G-estimation depends on some identifiability assumptions [37] which we describe below

## Conditional exchangeability and no measurement error

Like many causal models, our G-estimation requires the assumption of conditional exchangeability between exposed and unexposed subjects given earlier exposure and confounders at each visit, also known as no unmeasured confounders. Even if investigators succeed to identify and collect sufficient data on potential confounders using their expert knowledge, this assumption cannot be empirically tested [59]. In our study, collecting the 3-month average of time-varying exposure and confounders may result in residual confounding bias that violates exchangeability assumption. Moreover, measurement error of confounders such as serum albumin, ferritin, and white blood cell can arise residual confounding bias. Also measurement bias would occur due to binary classification of our continuous exposure. Although G-estimation has been extended for continuous exposure [60], its detailed application has been clarified just for a binary exposure [34, 42, 61].

### Well-defined intervention

This assumption is required for consistency i.e., for each subject, the counterfactual survival time under the observed value of exposure is equal to the observed survival time [62, 63]., Since there are multiple versions of intervention to change MCI values, including eating rich dietary protein intake, exercising, or treating inflammation that may correspond to different causal effects on outcome, the causal interpretation of MCI-mortality relationship is not straightforward and must be made cautiously. However, it would be a simple monitoring index which triggers additional diagnostic and therapeutic steps.

### Positivity

This assumption indicates observing both exposed and unexposed subjects within each stratum of confounders [41]. Interestingly, in contrast to other causal methods [64] such as inverse-probability-of treatment weighting [25, 65–70] and g-formula [71–73], G-estimation results in an unbiased estimate even when positivity assumption violated [41], based on extrapolation to the empty cells assuming that no confounders are effect modifier [5, 36].

### Model specification

Both SAFTM and pooled logistic regression model should be correctly specified. However, the parameters estimated using G-estimation in a SAFTM are more robust to model misspecification than those generated by maximum likelihood of associational AFT Weibull model, since the SAFTM is a semiparametric model and based on exposure modeling [22]. It is important to note that standard models require all these assumptions plus one more assumption: no time-varying confounder affected by prior exposure.

## Conclusion

Our G-estimation method adds new insight to the existing literature on the effect of MCI and all-cause mortality. Using G-estimation, we have shown that declining lean body mass, defined by MCI, increases mortality in ESRD patients receiving hemodialysis which was substantially different from the results based on the standard models which generated biased effect estimates toward the null by mishandling the time-varying confounders. Therefore, it is recommended applying G-estimation as a more appropriate causal model in the presence of variables which have dual roles as confounders and mediators. It should be noted that inadequate sample size caused wide CI in our study.

## Acknowledgments

We would like to thank to the staff of hemodialysis centers for their helpful collaboration in both data collection and development of this research.

## Author Contributions

**Conceptualization:** Mohammad Aryaie, Hamid Sharifi, Mahyar Etminan, Mohammad Ali Mansournia.

**Formal analysis:** Mohammad Aryaie.

**Methodology:** Mohammad Aryaie, Mohammad Ali Mansournia.

**Software:** Mohammad Aryaie.

**Supervision:** Mohammad Ali Mansournia.

**Writing – original draft:** Mohammad Aryaie, Hamid Sharifi, Azadeh Saber, Farzaneh Salehi, Maryam Nazemipour, Mohammad Ali Mansournia.

**Writing – review & editing:** Farzaneh Salehi, Mahyar Etminan, Maryam Nazemipour, Mohammad Ali Mansournia.

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
