## [Decision Letter · Decision Letter 0]

30 Apr 2022

PONE-D-22-07490Longitudinal Causal Effect of Lean Body Mass on All-Cause Mortality in Patients with End-Stage Renal Disease: Accounting for Time-Varying Confounders Using G-estimation.PLOS ONE

Dear Dr. Mansournia,

Thank you for submitting your manuscript to PLOS ONE. After careful consideration, we feel that it has merit but does not fully meet PLOS ONE’s publication criteria as it currently stands. Therefore, we invite you to submit a revised version of the manuscript that addresses the points raised during the review process. Please provide a point-by-point response to reviewers, taking into consideration the following comments: 1-Please replace "lean body mass" by "modified creatinine index" in the title. 2-Please clarify what positive and negative values imply in this conclusion: Standard models demonstrated 9% (95% CI: -36% to +54%) and 16% (95% CI: -42% to +23%) shorter survival time in patients who were always in the decline MCI group than those who were always in no-decline MCI group throughout the follow-up. This effect was demonstrated to be 43% (95% confidence interval [95% CI]: -79% to -19%). 3-The reference 9 cited in the introduction addresses the general population and "low-grade inflammation", this should be clarified or reference removed. 4-As pointed by reviewer #2, transplantation is not a competing risk but rather considered as censored event in this study.

We look forward to receiving your revised manuscript.

Kind regards,

Mabel Aoun, MD, MPH

Academic Editor

PLOS ONE

Journal Requirements:

[NO authors have competing interests]. 

Reviewers' comments:

Reviewer's Responses to Questions

**Comments to the Author**

1. Is the manuscript technically sound, and do the data support the conclusions?

Reviewer #1: Yes

Reviewer #2: Yes

2. Has the statistical analysis been performed appropriately and rigorously? 

Reviewer #1: I Don't Know

Reviewer #2: Yes

3. Have the authors made all data underlying the findings in their manuscript fully available?

Reviewer #1: No

Reviewer #2: No

4. Is the manuscript presented in an intelligible fashion and written in standard English?

Reviewer #1: Yes

Reviewer #2: Yes

5. Review Comments to the Author

Reviewer #1: In this manuscript, Aryaie M and colleagues present the results of a retrospective study evaluating the association between lean body mass, as measured by the modified creatinine index, and mortality in a cohort of over 500 patients with end-stage kidney disease (ESKD) undergoing hemodialysis at three dialysis clinics in Iran. One of the main objectives of the study was to compare the results obtained using standard methods with those using G estimation. The authors conclude that the results of no association between lean body mass and mortality obtained using standard methods are biased. The manuscript is well written. The study question is well formulated and relevant. Below are a few comments for the authors.

1) The modified creatinine index (MCI) is not widely used, in part because requires the measurement of kt/v which is complex. One recommendation would be to include a second measure or proxy of lean body mass in the analysis.

2) The confidence intervals for all three estimates of the association between lean body mass and mortality (using standard methods or G estimation) are very wide. For example, using G-estimation, the authors report 43% (95% CI: -79% to -19%) shorter survival time in subjects with persistent decline in MCI group. This is an important limitation of the study. The authors should comments on this. Also, should 43% be -43%?

3) The authors should provide formal measurement of bias for both, standard methods and G estimation.

4) Please provide the definition of “MCI” decline (does it mean a negative slope? Or a negative value at each measurement compared with baseline?

5) It would be helpful to have a table comparing the demographic and clinical characteristics of patients who died vs those who did not.

6) Descriptive statistics regarding MCI are not provided.

7) In the discussion the authors mention a positive association between muscle mass and mortality. Perhaps they intended to say “low” muscle mass?

8) MCI is not spelled in the abstract.

Reviewer #2: Here is a list of specific comments. Note: there was no line number. Page numbering in reviews and comments is based on the line numbers in the Editorial Manager-generated PDF.

1. Page 2, Methods, 2nd paragraph:

(1a) I suggest clarifying what MCI represented.

(1b) I assumed the the time-varying exposure of interest was lean body mass. It was not clear why patients were dichotomized to the decline and no-decline groups.

2. Page 4, Study Population and Follow-up : I suggest including a sentence describing the baseline before the “follow-up” sentence.

3. Page 4, Study Population and Follow-up, 1st sentence: Please clarify if the patients were incident, prevalent or both patients.

4. Page 4, Exposure, Poential Confounders and Outcome: The exposure of interest was never defined in the Exposure, Potential Confounders and Outcome section.

5. Page 4, Exposure, Poential Confounders and Outcome, 1st sentence, “all visits”: I suggest describing the study design (e.g., number of visits in the Study Population and Follow-up section).

6. Page 4, Exposure, Poential Confounders and Outcome, 3rd sentence, “time-fixed”: Would it be more precise to call “time-fixed” as ‘baseline’?

7. Page 4, Exposure, Poential Confounders and Outcome, 6th sentence, “after exclusion . . . ”: I suggest including a patient flow diagram describing the selection of the patients.

8. Page 5, 1st paragraph, 2nd sentence, “A(t) indicates lean body mass at visit t”: Because the lean body mass was never used (MCI was used as its proxy), I suggest indicating MCI was used to replace lean body mass and stating the assumption that the causal effects from MCI were the same as the causal effects from lean body mass. In fact, the exposure of interest was the MCI status, not the MCI value.

9. Page 5, 1st paragraph, 7th sentence, “no arrows . . . ”: I thought this should be an assumption, not a fact.

10. Page 5, Standard Models, 1st sentence: Although the first model provided some information, it was not comparable to the G-estimation. I would consider removing it (but OK if you decide to keep it).

11. Page 5, G-Estimation:

(11a) I think the G-estimation section was important but difficult to digest for readers who are not familiar with causal inference terminology. Because the G-estimation approach was a better approach than the standard model with time-varying confounders (i.e., the aforementioned second model), I suggest, in Figure 1, depicting which paths cannot be adjusted using the standard model but the G-estimation can. For example, which paths in Figure 1 can be adjusted by the G-estimation but cannot by the standard models.

(11b) I suggest calling out the additional assumptions described in the Discussion section. These assumptions were required for the G-estimation approach to be a better approach.

12. Page 5, G-Estimation, 1st sentence, “causal parameters”: Please define the causal parameter first.

13. Page 5, G-Estimation, 1st sentence, “lean body mass”: MCI, not lean body mass, should be used from this point forward.

14. Page 6, 1st paragraph, 2nd sentence, “competing risk”: I suggest avoiding referring transplantation as a competing risk. The outcome was all-cause mortality. Technically, there would not be any competing event. Transplantation might be considered as a censoring event.

15. Page 6, Results, 1st paragraph, 1st sentence: Please add the patient flow diagram here (see Comment #7 above).

16. Page 6, Results, 2nd paragraph, 1st sentence, “MCI status”: Please define the MCI status in the Exposure, Potential Confounders and Outcome section.

17. Page 7, 1st paragraph, 2nd paragraph, “43% (95% CI: -79% to -19%)”: This was a strange presentation comparing to the numbers in Table 2.

18. Table 1: Because the MCI status was time-varying, please clarify in the manuscript the MCI status in Table 1 was measured at baseline. If not, what was it?

6. PLOS authors have the option to publish the peer review history of their article (what does this mean?). If published, this will include your full peer review and any attached files.

Reviewer #1: No

Reviewer #2: No

---

## [Author Response · Author response to Decision Letter 0]

15 Jun 2022

We appreciate the academic editor and reviewers for their deep and thorough review. We have revised our manuscript in the light of their useful suggestions and comments, and we hope our revision has improved the manuscript to a level of their satisfaction. 

Academic Editor

1-Please replace "lean body mass" by "modified creatinine index" in the title.

Thank you for this suggestion. It was replaced.

2-Please clarify what positive and negative values imply in this conclusion: Standard models demonstrated 9% (95% CI: -36% to +54%) and 16% (95% CI: -42% to +23%) shorter survival time in patients who were always in the decline MCI group than those who were always in no-decline MCI group throughout the follow-up. This effect was demonstrated to be 43% (95% confidence interval [95% CI]: -79% to -19%). 

Thank you for pointing this out. The negative values mean shorter survival, and the positive values mean higher survival. To clarify our sentences, we write “shorter survival” for 9%, 16%, and 43% after both standard and G-estimation estimates, which is equal to -9%, -16%, and -43%.

3-The reference 9 cited in the introduction addresses the general population and "low-grade inflammation", this should be clarified or reference removed.

The reference 9 was removed. 

4-As pointed by reviewer #2, transplantation is not a competing risk but rather considered as censored event in this study.

Thank you, it was corrected.

Comments to the Author

Review Comments to the Author

Reviewer #1: In this manuscript, Aryaie M and colleagues present the results of a retrospective study evaluating the association between lean body mass, as measured by the modified creatinine index, and mortality in a cohort of over 500 patients with end-stage kidney disease (ESKD) undergoing hemodialysis at three dialysis clinics in Iran. One of the main objectives of the study was to compare the results obtained using standard methods with those using G estimation. The authors conclude that the results of no association between lean body mass and mortality obtained using standard methods are biased. The manuscript is well written. The study question is well formulated and relevant. Below are a few comments for the authors.

1) The modified creatinine index (MCI) is not widely used, in part because requires the measurement of kt/v which is complex. One recommendation would be to include a second measure or proxy of lean body mass in the analysis.

Thank you for this suggestion. We agree the measurement of kt/v is a little bit complex; however, based on the literature, modified creatinine index (MCI), determined by sex, age, pre-dialysis serum creatinine, and single-pool Kt/V (spKt/V), could be a reliable and valid surrogate marker of lean body mass. 

2) The confidence intervals for all three estimates of the association between lean body mass and mortality (using standard methods or G estimation) are very wide. For example, using G-estimation, the authors report 43% (95% CI: -79% to -19%) shorter survival time in subjects with persistent decline in MCI group. This is an important limitation of the study. The authors should comments on this. Also, should 43% be -43%?

Thank you for pointing this out. In the Conclusion section, last line, we add the limitation of our study as follows:

It should be noted that inadequate sample size caused wide CI in our study. Moreover, 43% shorter survival is equal to -43%. 

3) The authors should provide formal measurement of bias for both, standard methods and G estimation.

Thank you for pointing this out. In the Discussion section, last line of paragraph 8, we also clarified our paragraph as follows:

It is important to note that standard models require all these assumptions (G-estimation assumptions) plus one more assumption: no time-varying confounder affected by prior exposure. 

4) Please provide the definition of “MCI” decline (does it mean a negative slope? Or a negative value at each measurement compared with baseline?

Thank you for this suggestion. We clarified it in the Method section, exposure, potential confounders and outcome part, lines 5-6 as follows:

According to changes in MCI in each visit compared to previous visit, patients were dichotomized to either the decline group or no-decline group.

5) It would be helpful to have a table comparing the demographic and clinical characteristics of patients who died vs those who did not.

Thank you for this suggestion. We added the comparison of the demographic and clinical characteristics of patients who died vs. those who were alive in Table 1.

6) Descriptive statistics regarding MCI are not provided.

Thank you, we have pointed this in Table 1. We also add the following sentence in Result section, paragraph2, line 2:

There were 297 patients in decline MCI group and 256 patients in no-declined MCI group. 

7) In the discussion the authors mention a positive association between muscle mass and mortality. Perhaps they intended to say “low” muscle mass?

Thank you for your attention. It was corrected. 

8) MCI is not spelled in the abstract.

Thank you, it was corrected.

Reviewer #2: Here is a list of specific comments. Note: there was no line number. Page numbering in reviews and comments is based on the line numbers in the Editorial Manager-generated PDF.

1. Page 2, Methods, 2nd paragraph:

(1a) I suggest clarifying what MCI represented.

Thank you for this suggestion. In the Introduction section, third paragraph, we have explained what MCI represent for as follows:

Modified creatinine index (MCI), determined by sex, age, pre-dialysis serum creatinine, and single-pool Kt/V (spKt/V), has been introduced as a reliable, valid, and simple surrogate marker of lean body mass. 

We also have defined it in the Method section, exposure, Potential Confounders and Outcome part.

(1b) I assumed the time-varying exposure of interest was lean body mass. It was not clear why patients were dichotomized to the decline and no-decline groups.

Thank you for your attention. According to changes in MCI (as a surrogate marker of lean body mass) in each visit compared to the previous visit, patients were dichotomized to either the decline group or no-decline group d for the following reason:

Although G-estimation has been extended for continuous exposure, its detailed application has been clarified just for a binary exposure. It has been described in Discussion section, last line of paragraph 5.

2. Page 4, Study Population and Follow-up : I suggest including a sentence describing the baseline before the “follow-up” sentence. 

Thank you for this suggestion. Baseline characteristics of the study population has been described in Table1. 

3. Page 4, Study Population and Follow-up, 1st sentence: Please clarify if the patients were incident, prevalent or both patients.

Thank you for pointing this out. The study population were incident ESRD patients, which has been clarified. 

4. Page 4, Exposure, Potential Confounders and Outcome: The exposure of interest was never defined in the Exposure, Potential Confounders and Outcome section.

Thank you for pointing this out. We have defined it in the mentioned section as follow:

MCI determined, by sex, age, pre-dialysis serum creatinine, and single-pool Kt/V (spKt/V), as a reliable, valid, and simple surrogate marker of lean body mass. 

5. Page 4, Exposure, Potential Confounders and Outcome, 1st sentence, “all visits”: I suggest describing the study design (e.g., number of visits in the Study Population and Follow-up section).

We have clarified the number of visits in the mentioned section as follows:

Based on expert opinion of a panel of nephrologists and epidemiologists, data on time-varying confounders were collected at all visits (0 to 34 with 3-month intervals)

6. Page 4, Exposure, Potential Confounders and Outcome, 3rd sentence, “time-fixed”: Would it be more precise to call “time-fixed” as ‘baseline’?

 Thank you for this suggestion. We have replaced it to time-fix or baseline confounders. 

7. Page 4, Exposure, Potential Confounders and Outcome, 6th sentence, “after exclusion . . . ”: I suggest including a patient flow diagram describing the selection of the patients.

Thank you. The number of missing was very low. So, instead of follow diagram, we have explained it in the manuscript as follow: 

Out of 568 patients with ESRD, 15 (2.6%) subjects with missing data at baseline or visit 1 were excluded. As a result, 553 ESRD patients were included in the study; 24 (4.3%) patients were censored during the follow-up: 4 due to loss to follow-up and 20 due to transplantation.

8. Page 5, 1st paragraph, 2nd sentence, “A(t) indicates lean body mass at visit t”: Because the lean body mass was never used (MCI was used as its proxy), I suggest indicating MCI was used to replace lean body mass and stating the assumption that the causal effects from MCI were the same as the causal effects from lean body mass. In fact, the exposure of interest was the MCI status, not the MCI value.

We have clarified our sentence as follows:

A(t) indicates MCI status as a surrogate measure of lean body mass at visit t. Moreover, the assumption that the causal effects from MCI were the same as the causal effects from lean body mass has been discussed in the Discussion section, well-defined intervention part. 

9. Page 5, 1st paragraph, 7th sentence, “no arrows . . . ”: I thought this should be an assumption, not a fact.

Thank you for your attention. We replace “no arrows . . . demonstrate” to “no arrows . . . assumes”

10. Page 5, Standard Models, 1st sentence: Although the first model provided some information, it was not comparable to the G-estimation. I would consider removing it (but OK if you decide to keep it).

Thank you for your attention. Unlike the second model, the first standard model is not subject to the bias conditioning on time-varying confounders which gives us some information comparing two standard models. 

11. Page 5, G-Estimation:

(11a) I think the G-estimation section was important but difficult to digest for readers who are not familiar with causal inference terminology. Because the G-estimation approach was a better approach than the standard model with time-varying confounders (i.e., the aforementioned second model), I suggest, in Figure 1, depicting which paths cannot be adjusted using the standard model but the G-estimation can. For example, which paths in Figure 1 can be adjusted by the G-estimation but cannot by the standard models.

Thank you for your attention. We have provided some explanation below the Figure 1 as follows:

Standard models are subject to two biases: over-adjustment bias (e.g., conditioning on L2 blocks the indirect effect of A1 on Y3 through L2), this bias occurs because L2 is a time-varying confounder affected by the exposure A1 as well as an unmeasured causal risk factors U2, and collider bias (e.g., conditioning on L2 is common effect of A1 and A2. So, conditioning on L2 associate A1 and U2, making A1 a non-causal risk factor Y3), this bias occurs because L2 is a time-varying confounder affected by prior exposure A1. But G-estimation appropriately account for such time-varying variables that can at times act as both mediators and confounder.

(11b) I suggest calling out the additional assumptions described in the Discussion section. These assumptions were required for the G-estimation approach to be a better approach.

Thank you for this suggestion. We have provided some explanation below the Figure 1. 

12. Page 5, G-Estimation, 1st sentence, “causal parameters”: Please define the causal parameter first.

Thank you. We have mentioned it in the parenthesis as follow:

The causal parameter (effect of MCI on all-cause mortality) with the counterfactual survival time….

13. Page 5, G-Estimation, 1st sentence, “lean body mass”: MCI, not lean body mass, should be used from this point forward.

Thank you. It was replaced.

14. Page 6, 1st paragraph, 2nd sentence, “competing risk”: I suggest avoiding referring transplantation as a competing risk. The outcome was all-cause mortality. Technically, there would not be any competing event. Transplantation might be considered as a censoring event.

Thank you. It was corrected. 

15. Page 6, Results, 1st paragraph, 1st sentence: Please add the patient flow diagram here (see Comment #7 above).

Thank you. The number of missing was very low. So, instead of follow diagram, we have explained it in the manuscript as follow: 

Out of 568 patients with ESRD, 15 (2.6%) subjects with missing data at baseline or visit 1 were excluded. As a result, 553 ESRD patients were included in the study; 24 (4.3%) patients were censored during the follow-up: 4 due to loss to follow-up and 20 due to transplantation.

16. Page 6, Results, 2nd paragraph, 1st sentence, “MCI status”: Please define the MCI status in the Exposure, Potential Confounders and Outcome section.

Thank you. It has been defined as follow:

MCI level determined, by sex, age, pre-dialysis serum creatinine, and single-pool Kt/V (spKt/V), as a reliable, valid, and simple surrogate marker of lean body mass, like other studies (12, 13). Then according to changes in MCI in each visit compared to the previous visit, patients were dichotomized to either the decline group or no-decline group. 

17. Page 7, 1st paragraph, 2nd paragraph, “43% (95% CI: -79% to -19%)”: This was a strange presentation comparing to the numbers in Table 2.

Thank you for pointing this out. We have indicated 43% shorter survival which is equal to 0.57 in table 2. 

18. Table 1: Because the MCI status was time-varying, please clarify in the manuscript the MCI status in Table 1 was measured at baseline. If not, what was it?

Thank you. It was corrected.

---

## [Decision Letter · Decision Letter 1]

11 Jul 2022

PONE-D-22-07490R1.Longitudinal Causal Effect of Modified Creatinine Index on All-Cause Mortality in Patients with End-Stage Renal Disease: Accounting for Time-Varying Confounders Using G-estimation.PLOS ONE

Dear Dr. Mansournia,

Thank you for submitting your manuscript to PLOS ONE. After careful consideration, we feel that it has merit but does not fully meet PLOS ONE’s publication criteria as it currently stands. Therefore, we invite you to submit a revised version of the manuscript that addresses the points raised during the review process.

The authors are kindly asked to address the concerns of Reviewer 1.

We look forward to receiving your revised manuscript.

Kind regards,

Mabel Aoun, MD, MPH

Academic Editor

PLOS ONE

Journal Requirements:

Reviewers' comments:

Reviewer's Responses to Questions

**Comments to the Author**

1. If the authors have adequately addressed your comments raised in a previous round of review and you feel that this manuscript is now acceptable for publication, you may indicate that here to bypass the “Comments to the Author” section, enter your conflict of interest statement in the “Confidential to Editor” section, and submit your "Accept" recommendation.

Reviewer #1: All comments have been addressed

Reviewer #2: All comments have been addressed

2. Is the manuscript technically sound, and do the data support the conclusions?

Reviewer #1: Yes

Reviewer #2: Yes

3. Has the statistical analysis been performed appropriately and rigorously? 

Reviewer #1: I Don't Know

Reviewer #2: Yes

4. Have the authors made all data underlying the findings in their manuscript fully available?

Reviewer #1: No

Reviewer #2: No

5. Is the manuscript presented in an intelligible fashion and written in standard English?

Reviewer #1: Yes

Reviewer #2: Yes

6. Review Comments to the Author

Reviewer #1: The revised manuscript addressed the majority of the reviewer's comments. However, there is still a need to modify the presentation of the results. In the statement presenting the results (abstract and results section, Page 7), the direction of the association of the main effect is positive while the 95% confidence interval is negative. Please rectify this.

"G-estimation of SAFTM denoted 43% (95% CI: -79% to -19%) shorter survival time in subjects who would have been always in decline MCI group than those who would have been always in no-decline MCI group throughout the follow-up".

Consider using the same ratios presented in Table 2 rather than converting the ratios into percentages.

Reviewer #2: (No Response)

7. PLOS authors have the option to publish the peer review history of their article (what does this mean?). If published, this will include your full peer review and any attached files.

Reviewer #1: No

Reviewer #2: No

---

## [Author Response · Author response to Decision Letter 1]

12 Jul 2022

We appreciate the academic editor and reviewers for their deep and thorough review. We have revised our manuscript in the light of their useful suggestions and comments, and we hope our revision has improved the manuscript to a level of their satisfaction. 

1. Reviewer #1: The revised manuscript addressed the majority of the reviewer's comments. However, there is still a need to modify the presentation of the results. In the statement presenting the results (abstract and results section, Page 7), the direction of the association of the main effect is positive while the 95% confidence interval is negative. Please rectify this.

"G-estimation of SAFTM denoted 43% (95% CI: -79% to -19%) shorter survival time in subjects who would have been always in decline MCI group than those who would have been always in no-decline MCI group throughout the follow-up".

Consider using the same ratios presented in Table 2 rather than converting the ratios into percentages.

Thank you for this suggestion. We have modified the presentation of the results according to the reviewer’s comments based on Table 2.

---

## [Editor Report · Decision Letter 2]

15 Jul 2022

.Longitudinal Causal Effect of Modified Creatinine Index on All-Cause Mortality in Patients with End-Stage Renal Disease: Accounting for Time-Varying Confounders Using G-estimation.

PONE-D-22-07490R2

Dear Dr. Mansournia,

We’re pleased to inform you that your manuscript has been judged scientifically suitable for publication and will be formally accepted for publication once it meets all outstanding technical requirements.

Kind regards,

Mabel Aoun, MD, MPH

Academic Editor

PLOS ONE
---

## [Editor Report · Acceptance letter]

10 Aug 2022

PONE-D-22-07490R2 

Longitudinal Causal Effect of Modified Creatinine Indexon All-Cause Mortality in Patients with End-Stage Renal Disease: Accounting for Time-Varying Confounders Using G-estimation. 

Dear Dr. Mansournia:

I'm pleased to inform you that your manuscript has been deemed suitable for publication in PLOS ONE. Congratulations! Your manuscript is now with our production department. 

Kind regards, 

on behalf of

Dr. Mabel Aoun 

Academic Editor

PLOS ONE